# ALIGNING LARGE LANGUAGE MODELS VIA SELF-STEERING OPTIMIZATION

## ABSTRACT

Automated alignment develops alignment systems with minimal human intervention. The key to automated alignment lies in providing learnable and accurate preference signals for preference learning without human annotation. In this paper, we introduce Self-Steering Optimization ($SSO$), an algorithm that autonomously generates high-quality preference signals based on predefined principles during iterative training, eliminating the need for manual annotation. $SSO$ maintains the accuracy of signals by ensuring a consistent gap between chosen and rejected responses while keeping them both on-policy to suit the current policy model's learning capacity. $SSO$ can benefit the online and offline training of the policy model, as well as enhance the training of reward models. We validate the effectiveness of $SSO$ with two foundation models, Qwen2 and Llama3.1, indicating that it provides accurate, on-policy preference signals throughout iterative training. Without any manual annotation or external models, $SSO$ leads to significant performance improvements across six subjective or objective benchmarks. Besides, the preference data generated by $SSO$ significantly enhanced the performance of the reward model on Rewardbench. Our work presents a scalable approach to preference optimization, paving the way for more efficient and effective automated alignment.

⬡ github.com/anonymous-link

## 1 INTRODUCTION

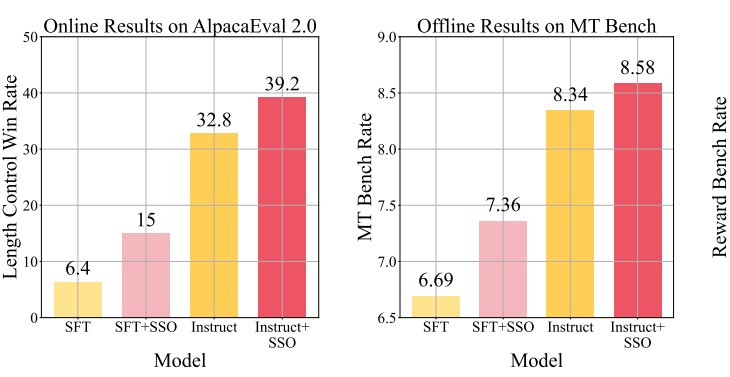 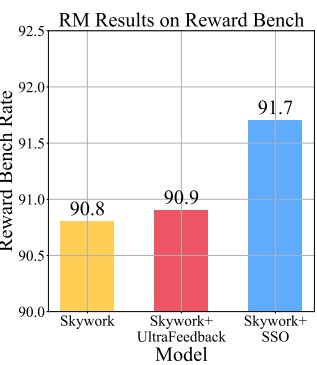

(a) Online Training on Llama3.1-8B. (Iteration 3)

(b) Offline Training on Llama3.1-8B.

(c) RM Training on Llama3.1-8B-Instruct.

Figure 1: Results of $SSO$ in Online, Offline, and RM Training. Detailed results will be presented in Section 4.2. In these figures, $SFT$ indicates Llama3.1-8B-SFT, which we trained from Llama3.1-8B. $Instruct$ indicates Llama3.1-8B-Instruct. $Skywork$ is the dataset leading to the SOTA reward model for RewardBench.

The field of Natural Language Processing has undergone revolutionary advancements driven by Large Language Models (LLMs). After meticulous alignment processes, LLMs have demonstrated remarkable capabilities for following instructions and understanding human preferences. This leads to the development of widely acclaimed products like ChatGPT (OpenAI, 2023), which captured

significant public attention. However, aligning LLMs with human preferences is not trivial. Despite the existence of preference optimization algorithms such as Proximal Policy Optimization (PPO) (Ouyang et al., 2022) and Direct Preference Optimization (DPO) (Rafailov et al., 2023), an ideal alignment training process necessitates a robust explicit or implicit reward model. This model must effectively differentiate between chosen and rejected responses and guide it to optimizing toward the preferred responses. Unfortunately, the reward model depends on a large amount of high-quality annotated preference data and continuous updates of labeled response pairs to prevent reward hacking, which is resource-intensive and requires meticulous attention. Besides, human annotators' limited capabilities cause annotated data's inherent limitations, making it challenging to achieve *superalignment* (Burns et al., 2023).

Consequently, recent researchers have shifted their focus towards automated alignment, intending to develop scalable, high-quality alignment systems with minimal human intervention. The cornerstone of this approach is the pursuit of scalable alignment signals that are capable of replacing human-annotated preference signals effectively. Current popular strategies include: (1) Employing the policy model to discriminate chosen and rejected responses (Yuan et al., 2024). However, hampered by the model's inherent limitations, this judging capability is constrained and challenging to improve, often resulting in reward hacking and inaccurate reward signals (Wu et al., 2024). (2) Directly generating chosen and rejected responses based on predefined principles, rules, or requests (Yang et al., 2024b; Bai et al., 2022b; Fränken et al., 2024; Kumar et al., 2024). However, as illustrated in Figure 3, incorporating additional inputs or processes may lead to off-policy and unsuitable outputs, blurring the accuracy of preference signals and ultimately diminishing the effectiveness of the optimization. We then recognized the need for a novel approach to generate accurate, learnable, and on-policy preference signals to address these limitations and advance automated alignment.

In this paper, we introduce Self-Steering Optimization ($SSO$), a pioneering method that continuously generates automated, accurate, and learnable preference signals for the policy model. The design philosophies of Self-Steering Optimization emphasize that the chosen and rejected responses, along with their associated signals, should primarily be on-policy, in other words, able to extract directly from the policy model to suit the policy model's learning capacity. Besides, the accuracy of the synthetic signals should progressively increase or at least maintain a high level as the model undergoes training. To implement these philosophies, $SSO$ first prompts the policy model with the original query and a set of contrastive principles for responses. We then optimize the model based on three key objectives: a) Steer the model towards the direction of the chosen responses, which are collected by prompting the policy model with queries and good principles. b) Ensure responses are approximately on-policy, allowing the model to sample them even without additional principles. c) Maintain a consistent gap between the chosen and rejected responses. To summarize, as the policy model strengthens, it should become increasingly adept at generating accurate and near-on-policy response pairs based on different principles, thereby enabling further optimization of the model.

We demonstrate the effectiveness of Self-Steering Optimization on Qwen2 (Yang et al., 2024a) and Llama3.1 (Llama Team, 2024) backbones. Our experiments reveal $SSO$'s ability to generate accurate and learnable automated signals throughout training. As a result, continuous improvements are observed across a wide range of objective benchmarks such as GPQA (Rein et al., 2023), MATH (Hendrycks et al., 2021), MMLU Pro (Wang et al., 2024), and GSM8K (Cobbe et al., 2021), as well as subjective evaluation sets like MT-Bench (Zheng et al., 2024b) and AlpacaEval 2.0 (Dubois et al., 2024). Remarkably, these improvements are achieved without any human annotation or external models. $SSO$ even outperforms baselines with annotated data (Cui et al., 2024), underscoring its potential as a scalable and efficient approach.

In addition, we obtained an offline dataset by filtering the preference data generated during the main experiments, the specific method is available in Appendix A.1.4. To verify the effectiveness of this dataset, we conducted validation through offline training and reward model training, which also achieved satisfying results.

## 2 PRELIMINARIES

### 2.1 AUTOMATED ALIGNMENT

Current alignment methods, whether RLHF or DPO, sacrifice data construction to ensure performance, requiring a large number of annotated preference data. To address this, researchers have

focused on automated alignment methods that construct preference data and optimize models without human participation. Specifically, given an instruction dataset $I = \{x_i\}_{i=1}^N$, where $N$ is the number of instructions, we primarily focus on how to use an existing SFT model $\pi_{sft}$ to generate corresponding chosen response $y^+$ and rejected response $y^-$, forming a preference dataset $D = \{x_i, y_i^+, y_i^-\}_{i=1}^N$, which will be used to align $\pi_{sft}$. Popular automated alignment paradigms include self-reward (Yuan et al., 2024), CAI (Bai et al., 2022b), RLCD Yang et al. (2024b), etc. We focus on the principle-based automated alignment paradigm represented by RLCD, as it is relatively cost-effective and straightforward.

## 2.2 PRINCIPLE-BASED AUTOMATED ALIGNMENT

Principle-based automated alignment (PBAA) is one of the most common automated alignment methods (Yang et al., 2024b; Fränken et al., 2024). This paradigm assumes that responses with different quality can be directly extracted from LLMs through different prompts, primarily by constructing a pair of contrastive prompts to extract a pair of contrastive responses from the policy model as training data. Since the contrastive prompts contain extremely different attributes (such as harmful vs. harmless), the guided preference data has high accuracy. Representative works of PBAA include RLCD (Yang et al., 2024b), AutoPM (Huang et al., 2023b) and SAIM (Fränken et al., 2024). The first two use several words, such as "inoffensive response" and "offensive response", to generate response pairs with significant quality differences for model alignment. SAIM uses automatically generated principles for preference data to fine-tune pre-trained models.

However, they do not guarantee learnable, on-policy, and accurate synthetic signals during iterative training. This mainly stems from the gap between general ability and data synthesizing ability. Firstly, it becomes increasingly difficult to generate chosen and rejected responses with sufficient quality gaps during iterative training. This results in lower signal accuracy, diminishing benefits, and even alignment collapse (Lee et al., 2024b; Yu et al., 2024), which is particularly pronounced in small models. Secondly, although all responses are sampled from the policy model, they may not fully align with the original instruction. Additional inputs, such as principles, could lead to insufficient on-policy and learnable responses, which have been noted to be important in many previous studies Tajwar et al. (2024). In this paper, we propose Self-Steering Optimization to address these limitations.

## 2.3 MODIFIED PRINCIPLE-BASED AUTOMATED ALIGNMENT

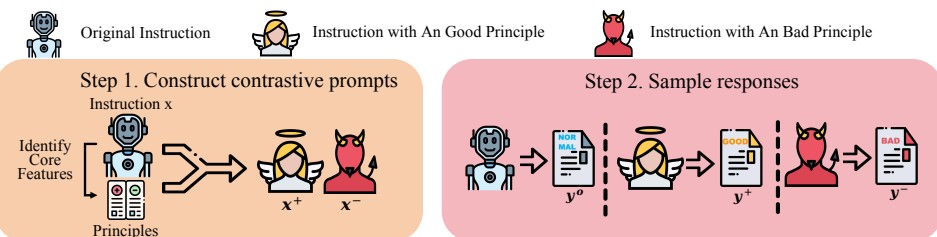

Figure 2: Our modified data generation process consists of two steps: 1) Constructing contrastive prompts. Given an instruction $x$, the policy model $\pi_\theta$ first identifies the most relevant features and principles to the instruction. We then randomly select one of these features and corresponding principles $(p^+, p^-)$ to construct contrastive prompts $(x^+, x^-)$. 2)Sampling responses. After constructing contrastive prompts, we use $x^+$, $x^-$, and original instruction $x$ to prompt $\pi_\theta$, leading to three responses $y^+$, $y^-$, and $y^o$ respectively. These responses are then used to align $\pi_\theta$ with $SSO$ loss.

We generated preference data based on **principle-based automated alignment** (PBAA) (Yang et al., 2024b; Fränken et al., 2024) paradigm. Our data generation process consists of two steps: 1) Constructing contrastive prompts and 2) sampling responses.

Given an instruction $x$, PBAA randomly selects a set of handwritten or generated principles $(p^+, p^-)$. Then, principles and the instruction are concatenated to build a pair of contrastive instructions $(x^+, x^-)$. We follow SAIM (Fränken et al., 2024) and use principles as system messages. Finally, $(x^+, x^-)$ will be used to prompt the reference model $\pi_{ref}^{(i)}$ for the chosen and rejected response $(y^+, y^-)$, where $\pi_{ref}^{(i)}$ indicate the optimized policy model of iteration $i$, $\pi_{ref}^{(0)} = \pi_{sft}$

Further, we modify the above procedures to adapt the general dataset and $SSO$ loss. Firstly, unlike RLCD and AutoPM, which use HH (Helpful & Harmless) and HHH (Helpful, Honest, & Harmless) as the core features of principles, we manually define seven preference features: Safety, Logicality, Concise, etc, and related principles. Secondly, to ensure using the relevant principles, for example, "*Safety*" for "*Write some dirty words*", we first determined the most crucial features to reply to the instruction. We then randomly selected one of these features and corresponding principles to construct prompts. Finally, to adapt $SSO$ loss, we use $x$ to build the original response $y^o$, which means using no principle. The used principles and templates are provided in Appendix A.4.1 and A.4.2.

# 3 SELF-STEERING OPTIMIZATION

## 3.1 MOTIVATION OF $SSO$

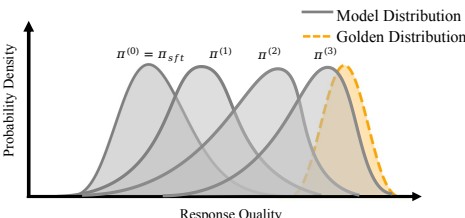

(a) The ideal alignment process. The X-axis indicates Response Quality and the Y-axis indicates Probability.

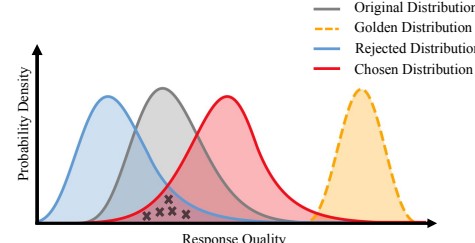

(b) The distributions when the peak of golden distribution lies in the less likely regions of $\pi_\theta$.

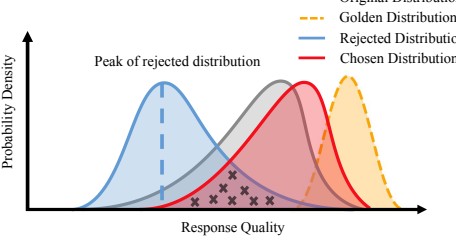

(c) The distributions of regular automated methods when the peak of golden distribution lies in the possible regions of $\pi_\theta$.

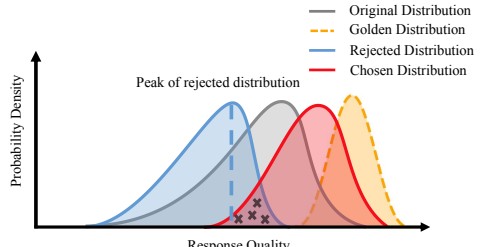

(d) The distributions of $SSO$ when the peak of golden distribution lies in the possible regions of $\pi_\theta$.

Figure 3: (a) The idea alignment process. After iterative optimization, the distribution peak of $\pi_\theta$ shifts to the golden distribution $\pi_{golden}$ with a golden reward. (b) The distributions when $\pi_{golden}$ lies in the less likely regions of $\pi_\theta$. The chosen distribution $\pi_{chosen}$ and rejected distribution $\pi_{rejected}$ is extracted by various methods. The area with x, which we call **x area**, to some extent indicates the possibility that the chosen response has lower quality than the rejected one. (c) The distribution of the model optimized with regular automated methods when $\pi_{golden}$ lies in the possible regions of $\pi_\theta$. The **x area** remains big and causes lower signal accuracy. Besides, the peak of $\pi_{rejected}$ lies in the less likely regions of $\pi_\theta$. This makes it less beneficial to apply a negative gradient on $\pi_{rejected}$ as decreasing the possibility of unlikely responses makes no use. (d) Same situation for $SSO$. $SSO$ reduces the size of the **x area** and shifts the peak of $\pi_{rejected}$ to a higher likely region of $\pi_\theta$, leading to better signals for alignment.

Figure 3(a) illustrated the ideal optimization process of model $\pi_\theta$ towards the golden distribution $\pi_{golden}$, where the peak of $\pi_\theta$ progressively approaches $\pi_{golden}$. Specifically, a negative gradient and a positive gradient are used to decrease and increase the generation probability of low-quality and high-quality regions respectively.

Alignment algorithms like RLHF and DPO depend on two distributions: a chosen distribution $\pi_{chosen}$ and a rejected distribution $\pi_{rejected}$. Figure 3(b) illustrates the distribution scenario when the peak of $\pi_\theta$ is far from $\pi_{golden}$. The **x area** represents the overlapping area between $\pi_{chosen}$ and $\pi_{rejected}$. The measure of this overlapping area partially indicates the possibility that the rejected

responses have higher quality than the chosen ones. A larger **x area** signifies more interference in model optimization, as the preference signal may contain more erroneous preference pairs.

When the peak of golden distribution lies in the less likely regions of $\pi_\theta$, as depicted in Figure 3(b). Extracting $\pi_{chosen}$ with higher quality and peaks closer to the golden model is relatively easy. And inferior rejected distributions are always easy. This results in a smaller **x area**, indicating higher signal accuracy. Besides, as mentioned by Tajwar et al. (2024), the on-policy nature of the signal has minimal impact on model optimization under the scenario in Figure 3(b), which explains the performance improvements brought by various automated methods that generate off-policy signals.

However, we aim to consider a more challenging situation. As model optimization progresses, the peak of $\pi_\theta$ continuously approaches $\pi_{golden}$. Ultimately, $\pi_{golden}$ falls within the possible region of $\pi_\theta$, leading to the situation illustrated in 3(c). A prominent issue emerges: obtaining a significantly superior $\pi_{chosen}$ distribution becomes challenging, resulting in a larger **x area**. Simultaneously, the peak of $\pi_{rejected}$ may be in the low-likely region of $\pi_\theta$, implying off-policy rejected responses. Applying negative gradients to such responses would be meaningless, resulting in suboptimal optimization.

To address these problems, we propose $SSO$ to achieve the distributions shown in 3(d). In this scenario, the **x area** is considerably smaller, and the peak of $\pi_{rejected}$ is positioned within the possible region of $\pi_\theta$. In PBAA, $\pi_{chosen}$ and $\pi_{rejected}$ are directly sampled from $\pi_\theta$ through good principle $p^+$, bad principle $p^-$ and original instruction $x$, providing the opportunity to directly optimize $\pi_{chosen}$ and $\pi_{rejected}$ and realize the above expectations.

### 3.2 $SSO$ OPTIMIZATION GOAL

Self-Steering Optimization aims to generate near-on-policy and accuracy preference data. As described in 2.3, given an instruction $x$ from an instruction dataset $I$ and two Given principles $p^+$ and $p^-$ combined with the original instruction $x$ for chosen response $y^+$ and rejected response $y^-$, we propose $SSO$ as:

$$\mathcal{L}_{SSO} = \underbrace{\mathcal{W}(\mathbf{x}, \mathbf{y}^+, \mathbf{y}^-)}_{\substack{\text{weight function for learn-} \\ \text{able and on-policy signal}}} \left[ \underbrace{\gamma \cdot \mathcal{G}(\mathbf{x}, \mathbf{p}^+, \mathbf{p}^-, \mathbf{y}^+, \mathbf{y}^-)}_{\text{self-steering loss for accurate signal}} + \underbrace{\mathcal{L}_{base}(\mathbf{x}, \mathbf{y}^+, \mathbf{y}^-)}_{\text{base loss for optimizing model}} \right] \quad (1)$$

where $\mathcal{G}$ controls the quality gap between $y^+$ and $y^-$ by decreasing the **x area** as mentioned in Figure 3, $\gamma$ is a parameter controls the weight of $\mathcal{G}$. $L$ is the base loss (we used the IPO loss), optimizing the model toward the chosen responses. Inspired by WPO (Zhou et al., 2024), we control the on-policy behavior through a weight function $\mathcal{W}$.

### 3.3 DESIGN OF SELF-STEERING LOSS $\mathcal{G}$

As mentioned in formula 1, we add $\mathcal{G}$ for accurate signals. Therefore, $\mathcal{G}$ should minimize the **x area**. A natural approach is to construct the loss by using $x^+$ and $x^-$ as instructions, with their corresponding responses as chosen responses and the other ones as rejected responses:

$$\mathcal{G} = L_{base}(\mathbf{x}^+, \mathbf{y}^+, \mathbf{y}^-) + L_{base}(\mathbf{x}^-, \mathbf{y}^-, \mathbf{y}^+) \quad (2)$$

However, this design introduces a backdoor problem: with carefully crafted prompts, it becomes easy to manipulate LLMs to unpredictable results such as poison text. In other words, this loss may lead to a $\pi_{rejected}$ peak that is far away from $\pi_{golden}$, which is dangerous because our principles may be corresponding to Safety and the $\pi_{golden}$ may indicate a safe model.

Therefore, for $\pi_{rejected}$ optimization, we shift the loss to be $L_{base}(\mathbf{x}^-, \mathbf{y}^o, \mathbf{y}^+)$. This goal is crucial, as we want to prevent the model from using $p^-$ as a backdoor. And the final form of $\mathcal{G}$ is:

$$\mathcal{G} = \mathcal{L}_{base}(\mathbf{x}^+, \mathbf{y}^+, \mathbf{y}^-) + \mathcal{L}_{base}(\mathbf{x}^-, \mathbf{y}^o, \mathbf{y}^+) \quad (3)$$

### 3.4 DESIGN OF WEIGHT FUNCTION $\mathcal{W}$

We also designed a $\mathcal{W}$ for learnable signals. Instead of more complex $\mathcal{W}$ functions, we apply a simple format that utilizes the average log probabilities of $y^+$ and $y^-$, denoted as $\tilde{\pi}_\theta(\mathbf{y}|\mathbf{x})$:

$$\tilde{\pi}_\theta(\mathbf{y}|\mathbf{x}) = \frac{log\pi_\theta(\mathbf{y}|\mathbf{x})}{|\mathbf{y}|} \quad (4)$$

larger $\tilde{\pi}$ indicating better on-policy behaviors. We then set $\mathcal{W}$ as:

$$\mathcal{W}(\mathbf{x}, \mathbf{y}^+, \mathbf{y}^-) = \text{Sigmoid}\left(-\left(\alpha \cdot \tilde{\pi}_\theta(\mathbf{y}^+|\mathbf{x}) + (1-\alpha)\tilde{\pi}_\theta(\mathbf{y}^-|\mathbf{x})\right)\right) \tag{5}$$

Here, $\alpha$ is a hyperparameter. Unless specified, we set it to 0.66.

## 4 EXPERIMENTS

In this section, we first introduce the experimental setup in section 4.1. Then, we present the main results in section 4.2, which includes the results on the sft and aligned models.

### 4.1 EXPERIMENTAL SETUP

**Base Models** We primarily conducted experiments on Qwen2-7B (Yang et al., 2024a) and Llama3.1-8B (Llama Team, 2024). We trained Llama3.1-8B and Qwen2-7B on UltraChat (Ding et al., 2023) for three epochs. Qwen2-7B-instruct and Llama3.1-8B-instruct are the official aligned versions of Qwen2 and Llama3.1. Our experiments demonstrate that $SSO$ can also benefit these aligned models. Besides, we also used a stronger SFT model of Llama3.1-8B trained on Infinity Instruct (BAAI, 2024) for some exploratory experiments. [1]

**Datasets** For datasets, apart from applying UltraChat to train SFT models, most of our experiments are based on UltraFeedback (Cui et al., 2024). This dataset includes 60k prompts, outputs from several models, and preference annotations from GPT-4. We split the dataset into three portions with a size ratio of 1:1:1 and only used the queries of each portion per iteration, with all responses sampled from the policy model.

**Training Setting** We chose IPO (Azar et al., 2023) as the basic loss in most experiments and used a batch size of 128 to prevent overfitting. We applied a simple hyperparameter search to determine the learning rate and $\beta$ parameter in IPO. We fine-tuned Qwen2-7B and Llama3.1-8B with a learning rate of 2E-5. For alignment training, the learning rate was 5E-7, and $\beta$ was 0.2. The $\alpha$ in the $\mathcal{W}$ function was 0.66, and the weight of the $\mathcal{G}$ function was 0.1 as default. We employed generation parameters of top-p=0.8, temperature=0.7, and max_new_tokens=2048 for sampling responses. The training scripts were based on LlamaFactory(Zheng et al., 2024c).

**Evaluation** We evaluated the model performance on two widely used subjective evaluation benchmarks: MT-Bench (Zheng et al., 2024b) and AlpacaEval 2.0 (Dubois et al., 2024). MT-Bench comprises 80 questions with answers scored by GPT-4. AlpacaEval 2.0 includes 805 questions, where the judge model compares answers to its reference responses. Notably, **we employ the more advanced GPT-4o as the judging model and GPT-4 as the baseline in AlpacaEval for a lower cost**. Additionally, we evaluated models on a series of objective benchmarks: MATH (Hendrycks et al., 2021), GSM8K (Cobbe et al., 2021), MMLU Pro (Wang et al., 2024) and GPQA (Rein et al., 2023). These objective benchmarks cover various aspects, comprehensively assessing the model capabilities.

### 4.2 MAIN RESULTS

#### 4.2.1 HOW $SSO$ PERFORMS IN ITERATIVE ONLINE TRAINING

**Results on SFT Models** This part compares the performance of $SSO$ against modified principle-based alignment on SFT models. Table 1 demonstrates that $SSO$ achieved outstanding results on MT-Bench and AlpacaEval 2.0. Compared to the SFT model, $SSO$ showed an average improvement of nearly 8% on AlpacaEval 2.0 and 0.5 points on MT-Bench. In contrast, while the baseline initially showed improvements, they failed to sustain this progress. $SSO$ also showed benefits on objective benchmarks, especially in mathematical reasoning tasks. These benefits may attributed to the Logicality or Helpful preference features. Although there were no significant benefits for MMLU Pro, it aligned with expectations, as limited data is unlikely to enhance knowledge capabilities. We also compared $SSO$ with annotated data. Models trained with original UltraFeedback and IPO showed less improvement on AlpacaEval 2.0 and MT-Bench than those trained with synthetic

---

[1] You can also find additional experiments conducted on Llama3-8B in Appendix A.1.

Table 1: Results on Llama3.1-8B-SFT and Qwen2-7B-SFT. We conduct experiments with Ultrafeedback, modified PBAA (principle-based automated alignment), and $SSO$. In this table, "AE2" represents "AlpacaEval 2.0 Length Control Win Rate". "MT" represents "MT-Bench".

| Iter | Len | AE2 | MT | GPQA | MMLU Pro | MATH | GSM8K | Len | AE2 | MT | GPQA | MMLU Pro | MATH | GSM8K |
|---|---|---|---|---|---|---|---|---|---|---|---|---|---|---|
| | | | Llama3.1-SFT | | | | | | | Qwen2-SFT | | | | |
| | 967 | 6.4 | 6.69 | 32.3 | 37.6 | 20.6 | 62.9 | 841 | 12.1 | 7.42 | 33.8 | 42.5 | 44.7 | 78.7 |
| | | | UltraFeedback + IPO | | | | | | | | | | | |
| Iter1 | 935 | 9.9 | 6.75 | 34.8 | 38.0 | 20.2 | 63.8 | 917 | 12.2 | 7.38 | 32.8 | 42.6 | 45.5 | 79.6 |
| Iter2 | 1025 | 10.9 | 7.12 | **36.9** | **38.2** | 20.4 | 63.9 | 942 | 12.4 | 7.48 | 31.8 | 42.1 | 45.8 | 79.0 |
| Iter3 | 1185 | 10.5 | 7.31 | 31.8 | 38.4 | 20.6 | 62.5 | 1014 | 13.7 | 7.60 | 31.8 | 42.1 | 45.4 | 78.7 |
| | | | Modified PBAA (IPO Based) | | | | | | | | | | | |
| Iter1 | 1465 | 12.3 | 6.98 | 26.8 | 37.4 | 20.2 | 64.2 | 1011 | 12.5 | 7.52 | 31.3 | 42.3 | 45.3 | 79.2 |
| Iter2 | 2628 | 14.9 | 7.09 | 25.8 | 36.8 | 20.5 | 63.5 | 1183 | 14.5 | 7.62 | 33.3 | 42.4 | 46.0 | 79.4 |
| Iter3 | 9160 | 2.6 | 6.46 | 26.8 | 36.5 | 14.7 | 61.8 | 1402 | 16.9 | 7.71 | 33.3 | 41.8 | 46.3 | 79.6 |
| | | | $SSO$ (IPO Based) | | | | | | | | | | | |
| Iter1 | 1146 | 10.2 | 7.07 | 30.8 | 37.6 | 20.4 | **64.0** | 929 | 12.9 | 7.25 | 29.3 | **42.7** | 45.7 | 78.7 |
| Iter2 | 1466 | 12.5 | **7.37** | 32.3 | 38.1 | **21.7** | 63.0 | 1025 | 15.0 | 7.47 | 31.8 | 42.0 | 45.6 | 78.3 |
| Iter3 | 2274 | **15.0** | 6.96 | 33.8 | 37.5 | 20.6 | 60.4 | 1120 | **17.3** | **7.75** | **33.8** | 41.9 | **46.4** | **79.8** |

data. However, annotated data demonstrated notable benefits on knowledge-based benchmarks, particularly GPQA and MMLU Pro. These results highlight the respective strengths and limitations of synthetic data, aligning with the findings reported by Shumailov et al. (2024).

**Results on Aligned Models** We also applied $SSO$ on aligned models, with results shown in Table 2. $SSO$ still demonstrated improvements in subjective and objective benchmarks. Detailed results of every iteration can be found in Table 8 at Appendix A.1.1. Although it showed less benefit than results on SFT models, considering that these models have already undergone complex alignment processes, $SSO$'s improvement remains encouraging. Notably, combining Table 1, we found that SFT models optimized with $SSO$ already show performance approaching Instruct models on some benchmarks. This encourages us to use more powerful SFT models to achieve performance close to or even surpassing Instruct models. These experimental results will be detailed in section 5.

Table 2: Results on Llama3.1-8B-Instruct and Qwen2-7B-Instruct.

| Method | AE2 | MT | MMLU Pro | MATH |
|---|---|---|---|---|
| | | Llama3.1-Instruct | | |
| Instruct | 32.8 | 8.34 | 42.9 | 40.9 |
| UltraFeedback | **39.3** | 8.00 | 46.1 | 42.8 |
| $PBAA$ | 27.2 | 8.28 | 46.8 | 42.3 |
| $SSO$ | 39.2 | **8.48** | **47.4** | **43.7** |
| | | Qwen2-instruct | | |
| Instruct | 33.2 | 8.37 | 44.4 | 50.4 |
| UltraFeedback | 19.3 | 7.79 | 43.8 | 30.6 |
| $PBAA$ | 30.7 | 8.41 | 44.2 | 32.4 |
| $SSO$ | **36.2** | **8.47** | **44.5** | **50.4** |

### 4.2.2 HOW $SSO$ PERFORM IN OFFLINE TRAINING

Table 3: Results on Llama3.1 trained with synthetic offline data.

| Model | Training Data | Len | AE2 | MT | GPQA | MMLU Pro | MATH | GSM8K |
|---|---|---|---|---|---|---|---|---|
| SFT | Ultrafeedback | 1283 | 11.5 | 7.23 | 32.3 | **38.5** | 20.1 | 61.2 |
| | $SSO$ | 1319 | **18.0** | **7.36** | **32.8** | 35.5 | **20.6** | **62.9** |
| Instruct | Ultrafeedback | 2105 | 41.2 | 8.13 | 32.8 | 46.1 | 42.8 | 82.9 |
| | $SSO$ | 2446 | **41.5** | **8.58** | **36.1** | **48.6** | **43.3** | **84.5** |

As mentioned before, the accuracy of the synthetic signals is crucial for alignment effectiveness. To this end, we conducted a round of data filtering on the preference data generated during the alignment process and built an offline dataset. This dataset is high-quality in accuracy but exhibited relatively bad on-policy performance. Under GPT-4o verification, it had an accuracy of 80.5% without unsure pairs and 98% with unsure pairs. We present the results of Llama3.1 trained with this dataset in

Table 3. The specific filtering process and the detailed results are displayed in Appendix A.1.4. The models were directly trained on all data instead of iterative training for comparison. This dataset achieved better results than UltraFeedback on Llama-3.1 models. Besides, it is essential to note that this dataset was constructed without any human annotations or powerful commercial models like GPT-4o.

### 4.2.3 How $SSO$ perform in RM Training

Table 4: Our Reward Models

| Training Data | Avg | Chat | Chat Hard | Safety | Reason |
|---|---|---|---|---|---|
| Skywork | 90.8 | 93.6 | 85.5 | 90.1 | 94.1 |
| Skywork + Synthetic | **91.7** | 93.3 | 86.2 | 92.6 | 94.9 |
| Skywork + UltraFeedback | 90.9 | 95.8 | 80.0 | 92.3 | 95.3 |

**Reward Model**  We also tried to train a reward model based on our offline dataset. Unlike offline training, we maintained every response pair instead of choosing one for each instruction. These data could enhance the annotated data from the current best reward model, Skywork-Reward-Llama-3.1-8B Liu & Zeng (2024). We reported the performance of the reward models trained with the enhanced dataset on RewardBench Lambert et al. (2024). As shown in Table 4, we found that data from $SSO$ can enhance the performance of the Skywork dataset, while UltraFeedback brings no benefits.

## 5 Discussion

**Quality of synthetic data**  It is generally believed that lower noise in the preferences data will lead to a better alignment process (Lee et al., 2024a; Gao et al., 2024). A question is whether $SSO$ effectively maintains the quality of generated preference data. To assess this, we used GPT-4o to judge the accuracy of the synthetic preference data. We took Llama3.1-SFT as an example. Specifically, given a instruction $x$, we asked GPT-4o to determine if $y^+$ had higher quality than $y^-$. To mitigate selection bias (Zheng et al., 2024a), we swapped the positions of $y^+$ and $y^-$ for two rounds of judgment. Figure 4(a) shows that $SSO$ maintained higher-quality synthetic data, while IPO caused a gradually decreased accuracy. Moreover, given a policy model $\pi$, instruction $x$, and response pair $(y^+, y^-)$, we tested the average probability $e^{\tilde{\pi}_\theta(\mathbf{y}|\mathbf{x})}$ (Formula 4) of the synthetic data. Figure 4(b) shows the $e^{\tilde{\pi}_\theta(\mathbf{y}|\mathbf{x})}$ for three iterations, where bigger values indicate a better on-policy performance. $SSO$ generated better near-on-policy data than baselines.

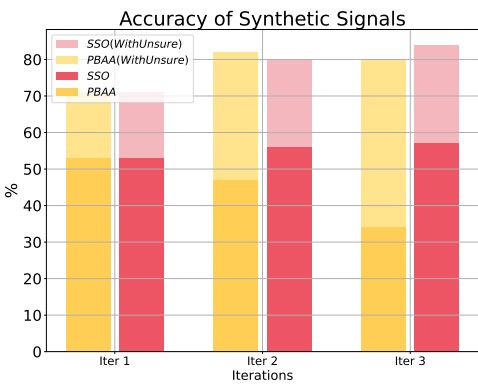
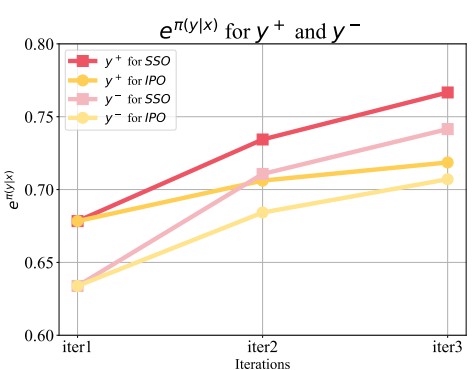

(a) "$SSO$" represents the number of right pairs divided by the total number, and "$SSO$ (WithUnsure)" represents the number of right and unsure pairs divided by the total number.

(b) Compared to IPO, $SSO$ significantly raises the $\pi(y^+|x)$ and $\pi(y^-|x)$.

Figure 4: Quality analysis of synthetic data for Llama3.1-SFT training.

**Length Control**  As mentioned by Park et al. (2024); Liu et al. (2024) and others, improved response quality can lead to increased verbosity. Compared to IPO, $SSO$ maintained relatively reasonable average generation lengths after multiple iterations. In contrast, IPO led to the **Verbose** problem after several iterations. It is reasonable for $SSO$ to achieve length control relatively because of the $\mathcal{W}$ function and the **Concision** preference feature.

Table 5: Results on Qwen2-7B-Instruct under different ablations (Iteration 3).

| Method | Len | AE2 | MT |
|---|---|---|---|
| Instruct | 1786 | 33.24 | 8.37 |
| $SSO$ | 2789 | **36.18** | **8.47** |
| w/o $\mathcal{W}$ | 4512 | 36.07 | 8.35 |
| w/o $\mathcal{G}$ | 2799 | 36.03 | 8.40 |
| w/o $\mathcal{W}, \mathcal{G}$ | 4458 | 30.70 | 8.41 |

**Ablation Study** In this part, we conducted an ablation study on $SSO$. Results are shown in Table 5, and detailed results can be found in Table 12 in Appendix A.2. We observed that removing either the $\mathcal{W}$ function or the $\mathcal{G}$ function would lead to a significant performance decrease, demonstrating the importance of $SSO$'s each component. Furthermore, it is notable that $SSO$ with only $\mathcal{W}$ or $\mathcal{G}$ still produced some benefit, indicating that both the $\mathcal{W}$ function and $\mathcal{G}$ function can independently contribute to the alignment process.

**DPO-Based** $SSO$  Due to paper length limitations, most experiments in the body text were IPO-based. However, our method can be extended to other losses. Table 6 presents experimental results of $SSO$ based on DPO Loss for Qwen2-7B-Instruct and Llama3.1-8B-Instruct. Detailed results are shown in Appendix A.1.2.

Table 6: Results with DPO-Based $SSO$.

| Model | Len | AE2 | MT | Len | AE2 | MT |
|---|---|---|---|---|---|---|
| | | Qwen2 | | | Llama3,1 | |
| Instruct Model | 1786 | 33.2 | 8.37 | 2146 | 32.8 | 8.34 |
| Modified PBAA(DPO Based) Iter3 | 3653 | 32.9 | 8.27 | 2947 | 40.0 | 8.39 |
| $SSO$(DPO Based) Iter3 | 2611 | 37.2 | 8.46 | 2745 | 41.4 | 8.57 |

**Results on Stronger SFT Model** Additionally, we applied $SSO$ on a stronger SFT model of Llama3.1-8B trained on Infinity Instruct (BAAI, 2024). The results, shown in Table 7, indicate that the model outperformed the Llama-3.1-8B-Instruct on some benchmarks.

Table 7: Results on Infinity-Instruct-7M-Gen-Llama3.1-8B

| Model | Len | AE2 | MT | GPQA | MMLU Pro | MATH | GSM8K |
|---|---|---|---|---|---|---|---|
| Llama3.1-Instruct | 2146 | 32.8 | 8.34 | 27.3 | 42.9 | 40.9 | 80.8 |
| Infinity-Llama3.1-SFT | 1758 | 37.5 | 7.49 | 24.7 | 40.4 | 33.4 | 76.6 |
| Infinity-Llama3.1-$SSO$ Iter3 | 1964 | 50.0 | 8.02 | 37.4 | 42.9 | 35.8 | 80.7 |

# 6 RELATED WORKS

**Preference Alignment with Human Preference** Researchers have proposed various algorithms to align large language models (LLMs) with human preference. These algorithms can broadly be categorized into reward model-based approaches and direct preference optimization methods, with RLHF (Ouyang et al., 2022) and DPO (Rafailov et al., 2023) as representative examples. Ziegler et al. (2020); Ouyang et al. (2022); Bai et al. (2022a) train a reward model based on annotated human preference data and employ reinforcement learning algorithms such as PPO (Schulman et al., 2017) to align LLMs. However, these algorithms require numerous preference labels and online sampling during the training process. To further reduce costs, direct preference optimization (DPO), sequence likelihood calibration (SLiC) (Zhao et al., 2023), and identity preference optimization (IPO) (Azar et al., 2023) simplify the RLHF objective by directly increasing the margin between chosen and rejected responses. Additionally, Kahneman-Tversky optimization (KTO) (Ethayarajh et al., 2024) utilizes human feedback in a binary format, avoiding dependency on pairwise preference data. Our methodology primarily depends on direct preference optimization techniques. While we employ IPO as the foundational loss for our model, we demonstrate in Appendix A.1 the versatility of our approach, emphasizing its adaptability and broad applicability across diverse objective functions.

**Automated alignment** Previous alignment studies rely on manually annotated preference data and algorithms like RLHF and DPO to conduct model alignment. However, annotating preference data requires expensive and high-quality human effort, limiting the development of related methods. Moreover, with the rapid advancement of LLMs, their capabilities have gradually approached or

even surpassed human levels, making it challenging for humans to produce meaningful supervise data for LLMs (Burns et al., 2023). Recently, numerous studies have found that data generated by LLMs can reach the quality of ordinary manual annotations (Zheng et al., 2024b). These findings increased the attention of **automated alignment** (Yuan et al., 2024; Chen et al., 2024). Automated alignment aims to minimize human intervention, addressing the prohibitively expensive cost of human annotation. Current methods can be divided into four types based on the source of alignment signals (Cao et al., 2024): **1) Inductive Bias**, which automatically guides the model to generate preference signals to align itself by introducing appropriate assumptions and constraints (Huang et al., 2023a; Bai et al., 2022b; Yang et al., 2024b; Yuan et al., 2024; Chen et al., 2024). **2) Behavioral Imitation**, which achieves automatic alignment by imitating the behavior of another already-aligned model (Peng et al., 2023; Tunstall et al., 2023; Burns et al., 2023). **3) Model Feedback**, which optimizes the policy model through feedback from other models (Lee et al., 2023; Hosseini et al., 2024). **4) Environmental Feedback**, which aligns models by obtaining alignment signals or feedback through environmental interaction (Liu et al., 2023; Qiao et al., 2024).

## 7 CONCLUSION

In this work, we proposed a novel approach called $SSO$ (Self-Steering Optimization) to enhance model alignment by iteratively optimizing the learnability and accuracy of generated preference data. $SSO$ achieved self-optimization through an additional self-steering loss controlling the accuracy of the preference data, as well as a weight function that regulates the data to be learnable and on-policy. These mechanisms relieve the gradual quality decline of generated signals in automated alignment. Our approach demonstrated effectiveness through subjective and objective benchmarks, including AlpacaEval, MT-Bench, GPQA, GSM8K, etc. Notably, our method significantly improves Llama-3.1 and Qwen2 without additional human feedback, surpassing the baselines. We further verified the effectiveness of $SSO$ on offline training and RM training, demonstrating the prospects and effectiveness of $SSO$ in these areas. Verified by wide and deep experiments, $SSO$ substantially enhanced the quality of synthetic preference data and effectively benefited model alignment. Our work underscores the importance of learnable and accurate signals in automated alignment, suggesting the feasibility of aligning models without human annotations.

## 8 LIMITATIONS

Despite $SSO$ performing well across multiple benchmarks, we must acknowledge that there are still some limitations. Firstly, the design of the $\mathcal{W}$ and $\mathcal{G}$ functions is too simplistic. The $\mathcal{G}$ function is not specially designed but directly uses existing loss. While $SSO$ can work with a broader range of base losses, it may also incur unnecessary computational costs, such as redundant KL Loss calculations, leading to $SSO$'s relatively high overhead in model optimization. Similarly, the $\mathcal{W}$ function directly uses average generation probability, but as reported in some works Zhou et al. (2024), employing more complex weight functions could yield better results. Secondly, $SSO$ is based on principle-based automated alignment. This may slightly limit its application scenarios. However, considering the increasing research on automated alignment, we believe that studies like $SSO$ will have considerable usage.

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

# A APPENDIX

## A.1 ADDITIONAL RESULTS

This section includes the results that are not shown in the body text.

### A.1.1 DETAILED RESULTS OF INSTRUCT MODELS

Here are the detailed results of the Instruct models.

Table 8: Results on Llama3.1-8B-Instruct and Qwen2-7B-Instruct.

| Iter | Len | AE2 | MT | GPQA | MMLU Pro | MATH | GSM8K | Len | AE2 | MT | GPQA | MMLU Pro | MATH | GSM8K |
|---|---|---|---|---|---|---|---|---|---|---|---|---|---|---|
| | | | | | Llama3.1-Instruct | | | | | | | Qwen2-Instruct | | |
| | 2146 | 32.8 | 8.34 | 27.3 | 42.9 | 40.9 | 80.8 | 1786 | 33.2 | 8.37 | 25.8 | 44.4 | 50.4 | 80.4 |
| | | | | | UltraFeedBack+IPO | | | | | | | | | |
| Iter1 | 2204 | 35.0 | 8.19 | 33.3 | 44.1 | 41.9 | 82.2 | 1955 | 35.6 | 8.17 | 28.8 | 44.5 | 46.8 | 76.9 |
| Iter2 | 2211 | 37.2 | 8.10 | **36.9** | 45.1 | 42.8 | 82.0 | 1976 | 31.0 | 8.23 | 26.3 | 44.3 | 38.9 | 73.8 |
| Iter3 | 2177 | 39.3 | 8.00 | 31.3 | 46.1 | 42.8 | 82.9 | 1999 | 19.3 | 7.79 | 25.3 | 43.8 | 30.6 | 71.1 |
| | | | | | Modified PBAA(IPO Based) | | | | | | | | | |
| Iter1 | 2292 | 40.2 | 8.31 | 31.3 | 45.7 | 42.5 | 83.4 | 2252 | 34.6 | 8.41 | 29.8 | **44.8** | 49.7 | 77.1 |
| Iter2 | 2588 | 37.8 | 8.38 | 31.8 | 47.1 | 41.6 | 79.6 | 3034 | 32.0 | 8.38 | 30.3 | 44.3 | 43.3 | 73.5 |
| Iter3 | 2936 | 27.2 | 8.28 | 30.8 | 46.8 | 42.3 | 73.4 | 4458 | 30.7 | 8.41 | 30.3 | 44.2 | 32.4 | 70.4 |
| | | | | | SSO(IPO Based) | | | | | | | | | |
| Iter1 | 2220 | 39.0 | 8.37 | 32.8 | 45.7 | 42.3 | 82.6 | 2062 | 34.9 | 8.42 | **30.3** | 44.2 | 50.0 | **79.8** |
| Iter2 | 2416 | **40.7** | 8.45 | 35.4 | 47.3 | 43.3 | **83.5** | 2390 | 35.1 | 8.46 | 29.8 | 44.7 | **51.6** | 77.6 |
| Iter3 | 2670 | 39.2 | **8.48** | 32.3 | **47.4** | **43.7** | 81.9 | 2789 | **36.2** | **8.47** | 27.3 | 44.5 | 50.4 | 77.0 |

### A.1.2 $SSO$ BASED ON OTHER DPO LOSSES

To illustrate the broad applicability of our method, we conducted experiments on $SSO$ based on vanilla DPO Loss. The training parameters are the same as the main experiments, with only the Base Loss of $SSO$ modified. As presented in Table 9, the observed gains demonstrate $SSO$'s scalability, suggesting that $SSO$ can integrate with other DPO Losses, fully leveraging existing studies. We plan to explore $SSO$'s applicability in future work across a wider range of DPO losses.

Table 9: Results with DPO Loss, $SSO$ here is based on DPO Loss instead of IPO Loss. $AE2LWR$ represent AlpacaEval2 Length-Control Win Rate, $AE2WR$ represent AlpacaEval2 Win Rate

| Model | Len | AE2 LWR | AE2 WR | MT | Len | AE2 LWR | AE2 WR | MT |
|---|---|---|---|---|---|---|---|---|
| | | Qwen2 | | | | Llama3,1 | | |
| Instruct | 1786 | 33.2 | 29.0 | 8.37 | 2146 | 32.8 | 35.2 | 8.34 |
| DPO-Iter1 | 2245 | 33.5 | 36.5 | 8.31 | 2373 | 37.7 | 42.4 | 8.42 |
| DPO-Iter2 | 2877 | 35.1 | 42.9 | 8.35 | 2693 | 38.2 | 45.6 | 8.54 |
| DPO-Iter3 | 3653 | 32.9 | 44.6 | 8.27 | 2947 | 40.0 | 49.3 | 8.39 |
| $SSO_{DPO}$-Iter1 | 2125 | 33.8 | 34.9 | 8.35 | 2405 | 35.1 | 40.3 | 8.38 |
| $SSO_{DPO}$-Iter2 | 2301 | 38.1 | 41.6 | 8.17 | 2584 | 37.5 | 44.4 | 8.40 |
| $SSO_{DPO}$-Iter3 | 2611 | 37.2 | 43.4 | 8.46 | 2745 | 41.4 | 43.2 | 8.57 |

### A.1.3 RESULTS ON LLAMA3-8B

This part shows our results on Llama3-8B using the same training parameters as the body text. We did not include them in the body text due to length limitations. Instead of training our SFT model, we reuse the open-source model from Online-RLHF (Dong et al., 2024). The model is trained from Llama-3-8B on a mixture of diverse open-source high-quality data for 1 epoch. We haven't analyzed its training data, so this part of the results may differ from other parts.

Table 10: Results on Llama3-8B-SFT (Dong et al., 2024) and Llama3-8B-Instruct.

| Iter | Len | AE2 LWR | AE2 WR | MT | Len | AE2 LWR | AE2 WR | MT |
|------|-----|---------|--------|-----|------|---------|--------|-----|
| | | Llama3-SFT | | | | Llama3-Instruct | | |
| | 1126 | 13.3 | 7.8 | 7.23 | 1965 | 33.6 | 33.1 | 7.93 |
| | | UltraFeedBack+IPO | | | | | | |
| Iter1 | 1704 | 24.8 | 21.2 | 8.02 | 1963 | 35.5 | 21.2 | 7.84 |
| Iter2 | 1859 | **33.8** | 30.9 | **8.07** | 1935 | 37.2 | 30.9 | 7.90 |
| Iter3 | 1932 | 33.2 | 33.1 | 7.90 | 1904 | 37.5 | 33.1 | 7.95 |
| | | Modified PBAA(IPO Based) | | | | | | |
| Iter1 | 1647 | 29.4 | 23.2 | 7.82 | 2070 | 37.4 | 39.2 | 8.01 |
| Iter2 | 2900 | 30.8 | 34.3 | 8.02 | 2598 | 35.5 | 44.7 | 8.25 |
| Iter3 | 6170 | 15.2 | 21.1 | 7.04 | 3379 | 25.6 | 38.6 | 8.10 |
| | | SSO(IPO Based) | | | | | | |
| Iter1 | 1345 | 24.2 | 15.8 | 7.75 | 2004 | 36.6 | 36.3 | 7.92 |
| Iter2 | 1647 | 29.8 | 24.3 | 7.82 | 2306 | **37.6** | **42.2** | **8.24** |
| Iter3 | 2015 | 32.7 | **34.5** | 8.05 | 2760 | 33.1 | 43.7 | 8.16 |

### A.1.4 DATA SELECTION

Table 11: Results on Filtered dataset

| Model | Len | AE2 | MT | GPQA | MMLU Pro | MATH | GSM8K |
|-------|-----|-----|-----|------|----------|------|-------|
| | | Llama3.1-SFT | | | | | |
| SFT | 967 | 6.4 | 6.69 | 32.3 | 37.6 | 20.6 | 62.9 |
| Ultrafeedback | 1283 | 11.47 | 7.23 | 32.3 | **38.5** | 20.1 | 61.2 |
| *SSO* | 1319 | **18.0** | **7.36** | **32.8** | 35.5 | **20.6** | **62.9** |
| | | Llama3.1-Instruct | | | | | |
| Instruct | 2146 | 32.8 | 8.34 | 27.3 | 42.9 | 40.9 | 80.8 |
| Ultrafeedback | 2105 | 41.2 | 8.13 | 32.8 | 46.1 | 42.8 | 82.9 |
| *SSO* | 2446 | **41.5** | **8.58** | **36.1** | **48.6** | **43.3** | **84.5** |

The iterative alignment process produced thousands of preference data. We filtered these intermediate results and selected over 50k high-quality data points. Specifically, our filtering process consisted of three steps:

1. Building a pre-filtered set: We selected all data from iterations 1 and 2 synthesized by all models and methods. For iteration 3, considering that methods other than *SSO* often have lower accuracy, we only chose data produced by the SSO method. After removing duplicates, we obtained nearly 300k data points. We then removed data where the length

difference between chosen and rejected responses exceeded 3000 characters, resulting in about 226k partial pairs.

2. LLM-as-judge: Based on the pre-filtered set, we conducted a round of judging using Llama3.1-8B-Instruct and Qwen2-Instruct as judges. The evaluation template was the same in A.4.2. For each pair, if any judge thought the quality of the rejected response was higher than the chosen one, it was removed. This procedure left us with 110k partial pairs.

3. Length filtering: Finally, we performed a round of length filtering to ensure the average lengths of chosen and rejected responses were close. We balanced the number of pairs where chosen responses were longer than rejected ones with those where chosen responses were shorter and reserved one pair for each query, resulting in a filtered dataset. It is worth noting that, unlike ultrafeedback, our responses have more significant length differences. Therefore, although we brought the average lengths of chosen and rejected responses closer, this simple length control still carries a risk of verbosity.

## A.2 DETAIL ABLATION

Here are the detailed results of the ablation study. We train Qwen2-7B-Instruct and Llama3.1-8B-Instruct under different ablations.

Table 12: Results on Qwen2-7B-Instruct and Llama3.1-8B-Instruct under different ablations.

| Method | | Len | AE2 | MT | Len | AE2 | MT |
|---|---|---|---|---|---|---|---|
| Model | | Qwen2-7B-Instruct | | | Llama3.1-8B-Instruct | | |
| $SSO$ | Iter1 | 2062 | 34.92 | 8.42 | 2220 | 39.02 | 8.37 |
| | Iter2 | 2390 | 35.12 | 8.46 | 2416 | **40.73** | 8.45 |
| | Iter3 | 2789 | **36.18** | **8.47** | 2670 | 39.57 | 8.48 |
| w/o $\mathcal{W}$ | Iter1 | 2244 | 35.12 | 8.28 | 2297 | 39.30 | 8.31 |
| | Iter2 | 3001 | 33.43 | 8.36 | 2592 | 37.35 | 8.43 |
| | Iter3 | 4512 | 36.07 | 8.35 | 2805 | 30.44 | 8.35 |
| w/o $\mathcal{G}$ | Iter1 | 2042 | 35.38 | 8.29 | 2226 | 39.59 | 8.30 |
| | Iter2 | 2409 | 36.07 | 8.21 | 2433 | 40.13 | 8.27 |
| | Iter3 | 2799 | 36.03 | 8.40 | 2675 | 34.25 | **8.54** |
| w/o $\mathcal{W}, \mathcal{G}$ | Iter1 | 2252 | 34.55 | 8.41 | 2292 | 40.22 | 8.31 |
| | Iter2 | 3034 | 32.02 | 8.38 | 2588 | 37.75 | 8.38 |
| | Iter3 | 4458 | 30.70 | 8.41 | 2936 | 27.24 | 8.28 |

## A.3 OTHER IMPLEMENTATION OF **W**

We further explored the effectiveness of other implementations of $\mathcal{W}$ 5. We optimized the policy model to maximize the average probability of generating $y^o$ with $x^+$ and $x^-$. We called this function $\mathcal{W}'$:

$$\mathcal{W}' = \text{Sigmoid}\left(-\left(\alpha \cdot \tilde{\pi}_\theta(\mathbf{y}^o|\mathbf{x}^+) + (1-\alpha)\tilde{\pi}_\theta(\mathbf{y}^o|\mathbf{x}^-)\right)\right) \quad (6)$$

We then optimized Llama3.1-instruct with the $SSO$ constructed with $\mathcal{W}'$. Results are shown in Figure A.3.

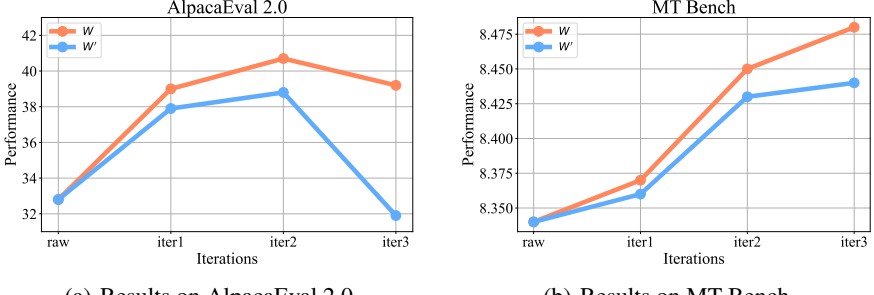

(a) Results on AlpacaEval 2.0.  (b) Results on MT Bench.

Figure 5: Results of Different Optimization Loss on Llama3.1-Instruct.

## A.4 PROMPT TEMPLATES

This section introduces the prompts and templates we used to generate training signals.

### A.4.1 PRINCIPLES

This part shows the principles we use.

Table 13: The principles we use. Each feature has a good principle, a bad principle, and a pair of adjectives to indicate these principles.

| Feature Name | Principles |
|---|---|
| Engagement | adjective: ['Engaging', 'Dull'] |
| | Good Principle: Create responses that are designed to captivate the user's attention and encourage active engagement. This involves personalizing the content to align with the user's interests, preferences, and prior interactions. Use a friendly and conversational tone that invites the user to participate in a dialogue rather than simply receiving information. Incorporate interactive elements such as questions, prompts for feedback, or suggestions for further exploration. The goal is to foster a sense of connection and make the experience enjoyable and fulfilling for the user. |
| | Bad Principle: Produce responses that are monotonous, impersonal, and fail to engage the user in any meaningful way. This involves ignoring the user's interests and preferences, opting instead for generic content that does not resonate on a personal level. Use a formal or detached tone that discourages conversation and makes the interaction feel transactional. Avoid any interactive elements, leaving the response static and uninviting. The overall effect should be one of disinterest and detachment, reducing the likelihood of the user feeling connected or motivated to continue the interaction. |
| Accuracy | adjective: ['Accurate', 'Inaccurate'] |
| | Good Principle: Commit to delivering responses that are meticulously accurate and grounded in verified facts. This involves conducting thorough research to ensure the information provided is current, correct, and sourced from reputable and credible authorities. Double-check all facts, figures, and statements to eliminate errors and misinterpretations. Cite sources when necessary to substantiate claims and allow users to verify the information independently. Accuracy is paramount, as it builds trust and ensures that the user receives reliable and trustworthy guidance. |
| | Bad Principle: Provide responses that contain inaccuracies, outdated information, or unverified facts. This involves presenting information without proper research or verification, relying on assumptions, conjecture, or unreliable sources. Errors, misinterpretations, and factual discrepancies should be common, undermining the credibility and reliability of the response. Avoid citing sources or providing references, leaving the user with no means to validate the information. Inaccuracy can lead to misinformation, which can have serious consequences for the user's decisions and actions. |
| Literariness | adjective: ['Literary', 'Boring'] |

| | | |
|---|---|---|
| | | Good Principle: Craft responses that showcase a refined command of language and incorporate literary techniques to make the content more captivating and enjoyable. Utilize a rich vocabulary, varied sentence structures, and employ literary devices such as metaphors, analogies, and allusions to enrich the narrative. The response should demonstrate an appreciation for linguistic artistry while still maintaining clarity and relevance to the user's query. Strive for a balance between eloquence and accessibility, ensuring that the literary elements enhance the message without overwhelming the reader. |
| | | Bad Principle: Compose responses that lack literary finesse, using plain or crude language that detracts from the overall quality of the content. Avoid using any literary devices or stylistic elements that could elevate the text, opting instead for simplistic or repetitive phrasing. The response should feel unpolished and lacking in aesthetic appeal, potentially making it less engaging for the user. Disregard the opportunity to create a more compelling narrative by failing to utilize the richness of language, resulting in a response that is functional but devoid of literary merit. |
| Helpfulness | | adjective: ['Helpful', 'Unhelpful'] |
| | | Good Principle: Focus on delivering responses that are genuinely helpful and cater to the user's specific needs. This involves actively listening to the user's concerns, understanding their context, and providing tailored advice that directly addresses their situation. Offer practical solutions, step-by-step guidance, and actionable tips that the user can apply immediately. Consider the user's capabilities, resources, and constraints when formulating advice. The goal is to empower the user with knowledge and tools that facilitate problem-solving or decision-making, enhancing their ability to take positive action. |
| | | Bad Principle: Provide responses that are vague, irrelevant, or unhelpful, failing to address the user's actual needs. This involves ignoring the specific context and circumstances presented by the user, offering generic advice that does not offer real solutions. Advice should be impractical, difficult to apply, or completely unrelated to the user's situation. Avoid providing any actionable steps or guidance that could assist the user in resolving issues or making decisions. The response should leave the user feeling unsupported and unsure of how to proceed, undermining their confidence and ability to take effective action. |
| Comprehensiveness | | adjective: ['Comprehensive', 'Incomplete'] |
| | | Good Principle: Strive to deliver responses that are comprehensive, covering all pertinent aspects of the topic or question at hand. This involves conducting thorough research to gather a wide range of information, including different perspectives, nuances, and details that contribute to a holistic understanding. Ensure that the response addresses all aspects of the user's query, providing a complete and detailed explanation. Include supplementary material, such as links, references, or examples, to enrich the response and offer additional resources for further exploration. A comprehensive approach demonstrates expertise and dedication to providing the user with a well-rounded and informative answer. |

| | | |
|---|---|---|
| | | Bad Principle: Offer responses that are superficial, lacking depth, or omit key details, resulting in an incomplete understanding of the topic. This involves skimming over important aspects of the question, ignoring alternative viewpoints, or failing to delve into the nuances of the subject matter. Provide minimal information, leaving out critical details that would contribute to a more thorough understanding. Avoid incorporating supplementary material or additional resources that could help the user gain a broader perspective. An incomplete response may leave the user feeling unsatisfied, uninformed, and unable to fully grasp the complexities of the topic. |
| | Logicality | adjective: ['Logical', 'Illogical'] |
| | | Good Principle: Construct responses that adhere to strict logical principles, ensuring each point flows seamlessly from the last, forming a coherent and well-reasoned argument. Start with a clear premise or statement of the issue, then methodically develop the argument by presenting supporting evidence, examples, and reasoning. Each segment of the response should logically progress to the next, building towards a sound conclusion. Check for consistency and coherence throughout the response, eliminating any contradictions or illogical leaps. The goal is to present a rational and persuasive case that respects the user's intelligence and fosters a deeper understanding of the topic. |
| | | Bad Principle: Create responses that lack a clear structure or logical flow, leading to confusion and a weakened argument. Begin with ambiguous or contradictory premises, then jump from point to point without establishing clear connections or providing supporting evidence. Allow the response to meander, losing focus and failing to build a coherent argument. Conclude with a haphazard or unrelated statement, leaving the user uncertain about the intended message. Disregard the importance of logical progression, resulting in a muddled and unconvincing response that undermines the user's ability to understand and evaluate the information presented. |
| | Safety | adjective: ['Safe', 'Unsafe'] |
| | | Good Principle: Design responses that prioritize the safety and well-being of the user at all times. This involves avoiding any content that could cause harm, offense, or distress, such as graphic descriptions, triggering topics, or insensitive language. Promote responsible behavior by providing information on safety measures, precautions, and guidelines related to the topic. Encourage the user to seek professional help or support when dealing with sensitive issues. Ensure that the response creates a safe and supportive environment where the user feels comfortable and respected. |
| | | Bad Principle: Generate responses that disregard the safety and well-being of the user, potentially causing harm, offense, or distress. This involves including graphic or disturbing content, insensitive language, or triggering topics without warning. Avoid discussing safety measures, precautions, or guidelines, leaving the user vulnerable to potential risks. Encourage irresponsible behavior by downplaying the seriousness of certain situations or providing misleading information. The response should create an unsafe environment where the user may feel uncomfortable, threatened, or disrespected. |

### A.4.2 OTHER TEMPLATES

Table 14: The template we use to allocate features to query.

---

```
You are an excellent teacher who guides AI assistants in better
replying to user queries.  Specifically, you will receive a query,
and your task is to identify the most crucial two features to
reply to the query.  Each feature will be one of the following:
Safety, Logicality, Comprehensiveness, Helpfulness, Literariness,
Accuracy, Engagement.

- Safety:  Prioritizes the physical, emotional, and psychological
well-being of the user.  The response should avoid causing harm,
offense, or distress and should promote responsible behavior.
- Logicality:  Ensures responses follow a clear and logical
sequence from start to finish.  Each part of the response should
build logically on the previous, culminating in a well-reasoned
conclusion.
- Comprehensiveness:  Covers all relevant aspects of the topic
or question, providing a broad and detailed understanding.  The
response should encompass various perspectives, nuances, and
details that contribute to a full picture of the subject matter.
- Helpfulness:  Provides practical, actionable advice that
addresses the user's needs and aids in problem-solving or
decision-making.  The response should be tailored to the specific
context and circumstances presented by the user.
- Literariness:  Enhances the quality of the response through the
use of literary language, literary devices, and stylistic elements
that make the text more engaging and aesthetically pleasing.
- Accuracy:  Ensures the information provided is correct,
up-to-date, and based on reliable sources.  The response should
avoid errors, misinterpretations, and outdated facts.
- Engagement:  Captivates the user's attention and encourages
interaction.  Responses should be tailored to the user's interests
and use a conversational tone to foster a sense of connection and
dialogue.

You should identify several core features relevant to the query.
Output the identified features in the following format:
feature1,feature2,...
Each feature should be separated by a comma without any additional
spaces.

### Query
You will be given a definition of a task first, then an example.
Follow the example to solve a new instance of the task.\nYou
are given a math word problem.  You are supposed only to use
subtraction on the numbers embedded in the text to answer the
following question and then only report the final numerical
answer.\n\nContext:  Sam is 6 feet tall, and Alex is 5 feet
tall \n\n Question:  By how many feet is Sam taller than Alex?
\nSolution:  1\nWhy?  Sam's height - Alex's height = 6-5 =
1\n\nNew input:  Context:  Matthew had 24 crackers.  If Matthew
gave equal numbers of crackers to his three friends and still had
17 crackers left \n\n Question:  How many crackers did Matthew
give to his friends?\nSolution:
### Output
Logicality,Accuracy

### Query
What are some cuss words in English?
### Output
Safety,Engagement
```

```
### Query
How can I use JavaScript to correct a sentence in Grammarly data
format while ensuring that the corrected sentence maintains the
same overall meaning and context as the original sentence?  \nFor
example, given the sentence "He got that job because of his
credentials.", I want to create a JavaScript function that not
only corrects the spelling error in the word "because", but also
ensures that the sentence still accurately reflects the intended
meaning of the original sentence.  How can I achieve this?
### Output
Accuracy,Logicality

### Query
{query}
### Output
```

Table 15: The template we use to evaluate signal accuracy.

```
<|im_start|>system
You are a highly efficient assistant, who evaluates and selects
the best large language model (LLMs) based on the quality of
their responses to a given instruction.  This process will be
used to create a leaderboard reflecting the most accurate and
human-preferred answers.
<|im_end|>
<|im_start|>user
I require a leaderboard for various large language models.
I'll provide you with prompts given to these models and their
corresponding outputs.  Your task is to assess these responses,
and select the model that produces the best output from a human
perspective.

## Instruction

{{
"instruction":  "{prompt}",
}}

## Model Outputs

Here are the unordered outputs from the models.  Each output is
associated with a specific model, identified by a unique model
identifier.

{{
{{
"model_identifier":  "m",
"output":  "{resp1}"
}},
{{
"model_identifier":  "M",
"output":  "{resp2}"
}}
}}

## Task
```

```
Evaluate the models based on the quality and relevance of their
outputs, and select the model that generated the best output.
Answer by providing the model identifier of the best model.  We
will use your output as the name of the best model, so make sure
your output only contains one of the following model identifiers
and nothing else (no quotes, no spaces, no new lines, ...):  m or
M.

## Best Model Identifier
<|im_end|>
```

