# OpenReview forum: "Aligning Large Language Models via Self-Steering Optimization"
_ICLR.cc/2025/Conference — ICLR 2025 Conference Withdrawn Submission_

### Official Review · Reviewer_e9XL · 2024-11-01

**Soundness:** 3
**Presentation:** 1
**Contribution:** 3
**Rating:** 3
**Confidence:** 4

**Summary:**

This paper introduces Self-Steering optimization, a preference finetuning method that automatically genreates perferences using contrastive pairs. SSO uses a combination of losses on automatically generated data to finetune an LLM.

**Strengths:**

* this paper tackles an important problem -- namely improving efficiency in the generation of alignment data.
* the paper does evaluation across a large number of benchmarks and sceniors (though the methodology and reasoning behind them is questionable, see weaknesses.)

**Weaknesses:**

**Writing**

The paper is a bit hard to approach without proper background, but htis is not provided by the paper. In several places notation is not proerly defined. See the "questions" section as well.

* I understand that the authors build on top of contrastive principles, but given an overview of this seems like necesary background.
* More intuition on the individual loss terms is necessary.
* Several grammar mistakes which should be corrected.

There are several unclear sentences / phrases in the paper. At present, I do not believe the writing passes the bar for publication. At the end of reading the paper, it is a bit unclear why the authors chose the specific losses / formulations used.

**Delta Versus Prior work**
It's unclear to me what the delta versus prior work is. Granted, I am not extermely familiar with principle based alignment. However, the authors do not do a good job articulating the differences between SSO and other methods. The closest they come to doing so is at the end of Section 2.1 where it is stated that "Additional inputs, such as principles, could lead to insufficient... we propose SSO to address these limitations"

What part of SSO is different than prior work? I assume prior work has the contrastive principle sampling? Is the difference then just the on-policy weighting function W? Why is this important? This also seems to be taken from WPO.


**Experiments**
The experiments section is not clearly written enough for me to discern what conclusions should be made. After reading the work, I was left with several questions about the methodology and presentation of the experiments:
* The Modified PBAA baseline is never defined.
* it doesn't make sense to me that the authors use ultra-feedback for training, but evaluate on MMLU-Pro and math. How does alignment influence math performance?
* Several of the results do not compare to baselines, and only present results for SSO. This includes Table 3 and Table 4

**Questions:**

Questions on teh writing in the draft:
* Several terms are not properly defined. What are principles $p^+$ ad $p^-$. Why are there only two of them?
* What is $y^0$ and where does it come from?
* How does $x^+$ relate to $p^+$.
* Several ambiguous terms. What does "accurate signal" mean?
* What does "We also designed a W for learnable signals" mean?

Questions on the method:
* Could the authors be precise about what the delta is versus prior work? I pose this question in more detail in the weaknesses section.

Questions on Experiemnts:
* The Modified PBAA baseline is never defined. What is it?
* Why do we evaluate alignment methods on benchmarks like MMLU-Pro and math? Looking at the appendix, the alignment principles often have nothing to do with these benchmarks, yet they are the core means of evaluation. How can we know how helpful SSO is for alignment if the reported benchmarks are not actually concerned with alignment.
* Why should we be able to compare PBAA-based methods and Ultrafeedback? It seems like these are just totally different datasets. Could the authors explain this?

---

### Official Review · Reviewer_EXrW · 2024-11-04

**Soundness:** 3
**Presentation:** 2
**Contribution:** 3
**Rating:** 5
**Confidence:** 3

**Summary:**

The paper introduces Self-Steering Optimization (SSO), an method used to align LLMs with minimal human intervention. SSO autonomously generates on-policy preference signals to guide the training of policy models without the need for manual annotation. This approach leverages predefined contrastive principles during iterative training to maintain a consistent quality gap between chosen and rejected responses. The paper validates SSO using two foundation models, Qwen2 and Llama3.1, showcasing significant performance gains across both subjective and objective benchmarks.

**Strengths:**

* The paper provides extensive benchmarking of the method and with additional experiments proving the robustness of the method.

* As the paper touched upon, the method can be extended to other loses that is not IPO-based which makes it more flexible.

* The method reduces reliance on costly human annotations, paving the way for more scalable training processes.

**Weaknesses:**

* The current design of the self-steering and weight functions is simplistic as mentioned in the limitations.

* The writing is a unclear at times and things in the method section could afford some more clarity. Especially reasoning about how your method solves your fundamental problem. Right now it's offered as a solution without going into details how.

* It's unclear what the author means with "Expectations" at section 2.3.

Overall, a plan on how you will improve the clarity of the introduction where you should clearly state the problem and then how your method mend this problem would go a long way.

**Questions:**

* How would your method scale with smaller models?

* How does SSO handle scenarios where human-like feedback is ambiguous or lacks clear contrastive principles?

Due to no responses before the deadline I am now lowering my score

---

### Official Review · Reviewer_nne3 · 2024-11-04

**Soundness:** 2
**Presentation:** 2
**Contribution:** 2
**Rating:** 3
**Confidence:** 4

**Summary:**

This paper introduces a novel method called Self-Steering Optimization (SSO) for automated large language models (LLMs) alignment. SSO autonomously generates high-quality preference signals based on predefined principles to aid in preference learning. The authors also propose a new optimization objective based on WPO and IPO. The method demonstrates effectiveness on Qwen2 and Llama3.1 models across multiple benchmarks compared to SFT models.

**Strengths:**

1. The automated alignment process reduces dependence on human annotations, making model alignment more scalable and cost-effective.
2. The results of SSO demonstrates improvements on benchmarks for both subjective and objective tasks with effective length control.
3. SSO can be extended to other base loss functions, e.g., IPO and DPO.

**Weaknesses:**

1. The improvement of this paper is mainly based on two categories, the synthetic data and the new objective. However, in the experiments the authors do not separate them well to state their effectiveness.
2. The clarity of this paper is not enough. The authors should provide more background on previous methods like WPO. The notations are also unclear. For example, $p^+, p^-$ from $\mathcal{G}$ defined in Equation (1) do not appear in following contents. Meanwhile, in Section 2.3, the authors introduce multiple QA pairs for their objective without well explaining their expectations.
3. The SFT baseline is based on basic data rather than the synthetic data. DPO/IPO with SSO data is also not compared.

**Questions:**

1. Can you show the improvement of SSO from the generative data and the proposed optimization objective separately?
2. Can you further explain why using $y^-$ in Equation (2) will cause a bookdoor problem? In Equation (3), why should $x^-$ prefer $y^O$ over $y^+$?
3. Why do you choose different base models in the experiments, e.g., the pretrained model, instruct model, and also SFT model (from Table 3)? Is the SFT model the baseline from previous experiments?
4. In Figure 4 (a), why can we see "IPO caused a gradually decreased accuracy" since both the optimization methods and the data are different?

---

### Official Review · Reviewer_RsyP · 2024-11-04

**Soundness:** 2
**Presentation:** 2
**Contribution:** 2
**Rating:** 3
**Confidence:** 2

**Summary:**

This paper proposes an auxiliary additive “self-steering” loss for iterative preference optimization algorithms (e.g. iterative IPO, DPO) for LLM alignment. This self-steering term is inspired from the principle-based alignment literature, and is designed to maintain a distinguishable gap between positive and negative responses despite sampling them on-policy.

**Strengths:**

This paper considers a very interesting idea of introducing principle-based methods into preference optimization algorithms such as DPO and IPO. Such methods, especially their iterative versions, have substantial drawbacks as identified by section 1 of this paper, and addressing them would go a long way in achieving scalable and efficient LLM alignment.

**Weaknesses:**

I found the paper to be weak in the following aspects:
1. **Experimental results.** Many of the improvements of the method seem very incremental, or within noise (Table 1). Seeing these results, I'm not convinced that this method offers noticeable improvements over existing baselines.
2. **Clarity.** The paper structure and writing were lacking in several areas (see below), and I found the method to be explained poorly despite its simplicity. In particular, the loss term could be explained and motivated much better in section 2.3.

**Questions:**

1. Related work (2.1) should be its own section preceding section 2.
2. Should not use theta for loss weight (since it’s commonly used to refer to policy parameters).
3. The problem setting is not clearly defined - should be defined in the beginning of section 2 or its own section.
4. Line 199/200 - what does this backdoor refer to? This needs to be more clearly explained.
5. No error bars are given in results. This is particularly because many of the results show little difference between SSO and the baselines.
6. GSM8K iter1 of SSO seems misbolded in Table 1 - it is lower than modified PBAA iteration 1.
7. I would argue all MATH and GSM8K (Table 1) results are within noise. AE2 is also marginal (15.0, vs 14.9 for PBAA iteration 2).
8. Understanding why PBAA AE2 drops significantly would be an interesting contribution.
9. A good ablation would be simply removing the self-steering term (and keeping the WPO-inspired term) to understand its impact.

---

### Note · Authors · 2024-12-16

I have read and agree with the venue's withdrawal policy on behalf of myself and my co-authors.